



# Evaluation of Soil Water Index of distributed Tank Model in a small basin with field data

Sofia Melo Vasconcellos [1], Masato Kobiyama [1], and Aline de Almeida Mota [2]

[1]Institute of Hydraulic Researches, Federal University of Rio Grande do Sul, Porto Alegre, Brazil
[2]Department of Environmental and Sanitary Engineering, Federal University of Fronteira Sul, Chapecó, Brazil

**Correspondence:** Sofia Melo Vasconcellos (sofia.vasconcellos@ufrgs.br)

**Abstract.** The objective of the present study was to determine the spatial behaviour of the Soil Water Index (SWI) by applying a distributed version of the Tank Model (D-Tank Model) to the Araponga river basin (5.26 ha) in southern Brazil and to verify its reliability through the comparison to soil moisture estimated with the measured water-tension values and the water retention curve. The study area has a monitoring system for rainfall, discharge (5- min interval), and soil-water tension (10-min interval).

The simulation results showed that the D-Tank Model has a reliable performance. The correlation between SWI and HAND was reasonable (r=0.6) meanwhile that between SWI and the Topographic Wetness Index was high (r=0.88). The comparison between the spatially distributed values of the SWI and soil moisture confirmed the high potential of the SWI derived from the D-Tank Model to be applied for predictions related to hydrological and environmental sciences. Keywords: D-Tank Model, Soil Water Index, Topographical Wetness Index, HAND, Soil Moisture.

## 1 Introduction

Soil moisture is very important in the energy and water exchanges between pedosphere and atmosphere, which exert direct influence on the processes of infiltration, drainage, evapotranspiration, surface runoff, among others (Entin et al., 2000). Therefore, soil moisture is widely used as a variable in a lot of environmental, hydrological, meteorological and agricultural studies (Walker et al., 2004). It is well known that the spatio-temporal monitoring of soil moisture supports better management of water

resources, prediction of floods and droughts, and so on. In spite of its importance, representing the spatial variation patterns of soil moisture is challenging (Mälicke et al., 2019), and this variable is not regularly monitored due to the high cost (Brocca et al., 2017).

To fill this gap, various methods for estimating soil moisture condition have been proposed. For example, the use of satellite measurements such as Soil Moisture and Ocean Salinity - SMOS (Kerr et al.,2001; Gumuzzio et al.,2016), indexes relating

soil moisture condition to antecedent precipitation such as the Antecedent Precipitation Index - API (Kohler and Linsley, 1951), and API-mod (Pellarin et al., 2013), indexes relating to soil moisture such as Soil Water Index (Okada et al.,2001;Paulik et al.,2014; Grillakis et al.,2016), Tank Moisture Index (Lindner and Kobiyama, 2009), Soil Moisture Index (Carrão et al., 2016), Soil Moisture Drought Index (Sohrabi et al., 2016), Soil Wetness Index (Saleem and Salvucci, 2002), and Soil moisture state (Zhuo and Han, 2017) and Topographical Wetness Index (TWI) (Beven and Kirkby, 1979) The relation between TWI and


soil moisture have been explored by several authors (Sørensen et al., 2006; Radula et al.,2018; Kim, 2012; Metzen et al.,2019). Similarly to the TWI, Rennó et al. (2008) found that the elevation difference to the nearest stream was correlated to soil water content distribution and proposed a new model, Heigh Above the Nearest Drainage (HAND). And both TWI and HAND are considered indicators of flood-prone areas (Nardi et al.,2006; Manfreda et al., 2015; Zheng et al.,2018; Tavares da Costa et al.,2019), and for landscape classification (Gharari et al.,2011; Loritz et al.,2019).

Physically-based distributed models have shown suitability for studying soil moisture and runoff response (Castillo et al.,2003; Loritz et al.,2019). Furthermore, some distributed hydrological models, such as WBMGB proposed by Saldanha et al. (2012) which divides the analyzed area into square cells of 10-km resolution, spatially calculate the water balance in the soil and consequently deal with soil moisture conditions. Sheikh et al. (2009) developed the BEACH model, a simple two-layer soil water balance model that provides spatially distributed initial soil moisture content. They emphasized the importance of soil

moisture in hydrological modeling. Rahmati et al. (2018) also concluded that it is important to evaluate the basin physical characteristics like soil moisture, in order to properly understand the hydrogeomorphic processes therein.

As soil moisture condition results from a water cycle, it is quite reasonable to treat it in the context of hydrological models simulating the water balance. One of the classic hydrological models is the Tank Model proposed by Sugawara (1961, 1995), which is originally lumped and deterministic. This model was well evaluated and recommended by World Metereological

Organization (WMO) (1975, 1992) and Franchini and Pacciani (1991). Since it is computationally simple and generates good results of hydrograph estimation, this model has been applied to various issues, for example, landslide prediction (Kobaishi and Suzuki,1987;Shuin et al.,2013; Nie et al.,2017); debris flow investigation (Takahashi and Nakagawa, 1991); flood forecasting (Tingsanchali and Gautam, 2000); Paddy field management (Chen et al., 2003); water quality (Kato, 2005); and sediment yield estimate (Lee and Singh, 2005), and also modified to its distributed version (Kato et al.,2005; Huang et al.,2007).

The Tank Model was initially considered a storage model, composed of a series of vertical reservoirs (tanks), which schematically represents the stratification of the soil layers from the surface to the base. The hypothesis based on this model structure is that the sum of the water storage calculated in these tanks would be a representative measure of the real condition of soil moisture in the basin. Then, the Soil Water Index and Tank Moisture Index derived from the Tank Model were proposed by Okada et al. (2001) and by Lindner and Kobiyama (2009), respectively. Thus, the use of the Tank Model as well as these indexes can

be very useful in hydrological studies and, consequently, in natural disasters, water resources and basin managements (Saito and Matsuyama,2012; Chen et al.,2013; Oku et al.,2014; Mukhlisin et al.,2015; Saito and Matsuyama,2015; Jun,2016; Chen et al.,2017;Matlan et al.,2018). However, the relationship between the water storage of each tank and the real soil moisture condition measured in a real basin has never been evaluated.

The objective of the present study was, therefore, to propose a distributed version of the Tank Model, to take into account the

model and parameter uncertainties to compare the Soil Water Index (SWI) of Okada et al. (2001) with soil moisture indicators (TWI and the HAND topology), and to compare the results with the soil moisture obtained in a small experimental basin.





## 2 Methods

### 2.1 Study Area

The study area is Araponga river basin (5.26 ha) which is a small experimental basin located in the rural area of Rio Negrinho

municipality of the Santa Catarina state, southern Brazil. In this basin, the mean slope of the main channel and drainage density are 0.30 $m.m^{-1}$ and 9.62 $\cdot 10^3 \cdot m^{-1}$ respectively (Mota et al., 2017). According to Alvares et al. (2013), the climate of the region is classified as Cfb - Humid subtropical, oceanic climate, without dry season, with temperate summer.

The basin is covered with the Mixed Ombrophilous Forest (Araucaria Forest or Brazilian Pine Forest) forming Montana (altitudes between 400 and 1000 m). The predominant soil in the region is the Cambisol, moderate and prominent with medium

and clayey texture. (Santa Catarina, 1986). From field survey, Mota (2017) identified two soil layers and denominated the top one as A and the subsurface layer as B. The layer A is 0.2-m thick and characterized as a dark brown soil with 78% sand, 18% silt, and 4% clay. The layer B is 1.2-m thick and characterized as a dark yellow soil with 19% sand, 28% silt, and 53% clay).

Figure 1(a) shows the hypsometric map of the basin and, also the location of the rain gauge, the discharge weir and 9 tensiometer-batteries. The rainfall and discharge data were automatically measured in 5-min interval and, soil water tension

data every 10 minutes.

According to Mota et al. (2017), the tensiometer sensors were installed by considering the soil depth and profile along the three hillslopes selected for the monitoring. In each hillslope, there were three points of tensiometer-batteries: near the river (higher slope with a mean soil depth of 0.5 m), at the middle hillslope (smaller slope with soil depths from 1.0 to 1.9 m), and near the basin divisor (smaller slope with soil depths from 1.0 to 1.9 m). Therefore, considering the variability of soil

depth along each hillslope, there were adopted two configurations for the depths' distribution of the soil water tension sensors. The type II configuration is characterized with the points very near the river (tensiometers A12, B12 and Z12), and the type I configuration with the other points (tensiometers A3, A4, B3, B4, Z3 and Z4). For the type II, 2 sets with 3 sensors were installed, at the 0.1 m, 0.3 m and 0.4 m depths. The type I was used at the points where the soil layer was deeper, which allowed the installation of 7 sensors at the depths 0.1 m, 0.2 m, 0.3 m, 0.4 m, 0.6 m, 0.75 m and 0.90 m. Their locations are shown in

Figure 1(b). The tensiometers used for the measurements were of the Irrometer Watermark, model 200SS.

### 2.2 Used Data

The topographic information of the basin was extracted from the digital elevation model (DEM) of the Araponga basin at 1:5000 scale of 1-m resolution that was generated through the topographic survey carried out by Mota et al. (2017).

The precipitation and discharge data used in the present study were obtained during the period from March 2011 to December

2015 as showed in Figure 2. The evapotranspiration was calculated from the measurements at the Feio meteorological station, located 3 km distant from the Araponga basin, using the same procedure of Lindner and Kobiyama (2009) which utilized the Penman modified method of Doorenbos and Pruitt (1977).

By using the equation of Van Genuchten (1980), Mota (2017) estimated the parameters of the water retention curves for the two soil-layers of the study area from 12 undisturbed soil samples (6 at 10-cm depth and 6 at 50-cm depth) collected near





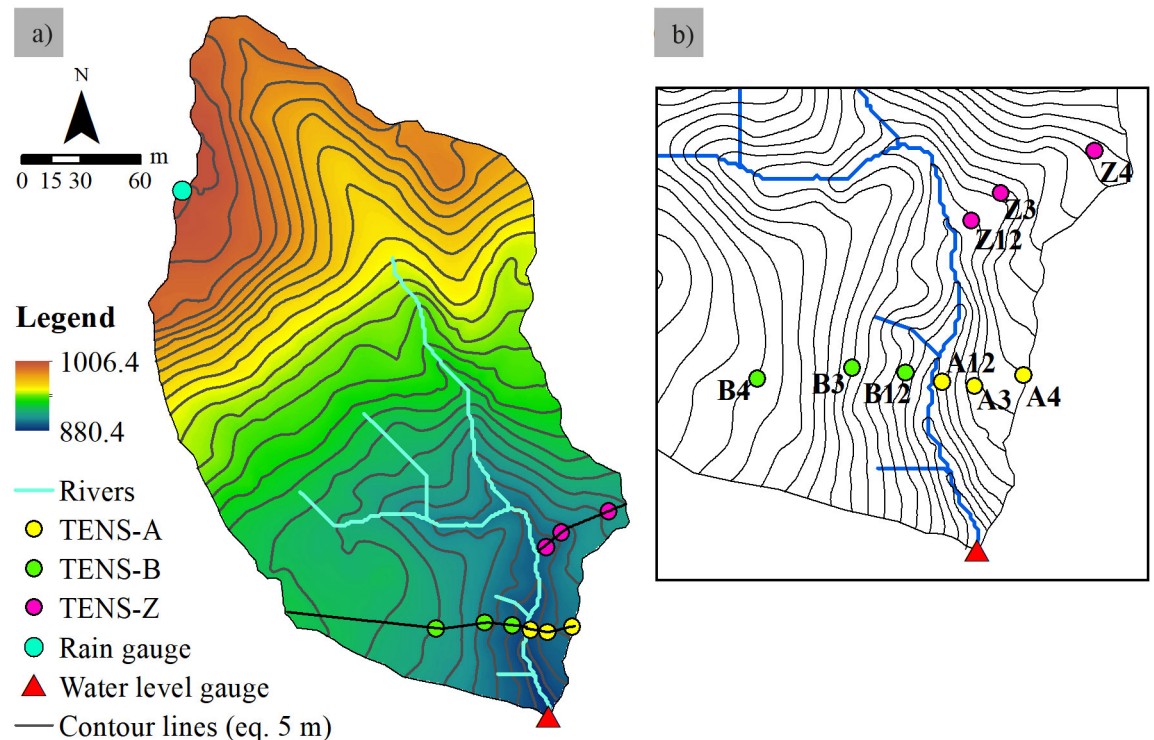

**Figure 1.** (a) Hypsometric map of the experimental Araponga river basin and location of the monitoring devices installed (Source: (Mota et al., 2017)), (b) Detail of the location of the tensiometers.

**Table 1.** Parameters of the water retention curves. Source: Mota (2017)

|  | Layer A | Layer B |
| --- | --- | --- |
| $\Theta_r (cm^3/cm^3)$ | 0.0896 | 0.3773 |
| $\Theta_s (cm^3/cm^3)$ | 0.6511 | 0.5518 |
| $K_s (cm/s)$ | $4.05 \cdot 10^3$ | $4.63 \cdot 10^4$ |
| $\alpha (cm^{-1})$ | 0.0004 | 0.0011 |
| n | 1.6165 | 3.6782 |
| m | 0.3814 | 0.7281 |

the installed tensiometers. These samples were subjected to predefined pressures (500, 1000, 3000, 6000 and 10000 cm) in the Richards pressure plate until the drainage stabilized and then their water contents were determined. The RETC program (Van Genuchten et al., 1991) was used to fit the retention curves from pressure head and soil water content (Table 1).

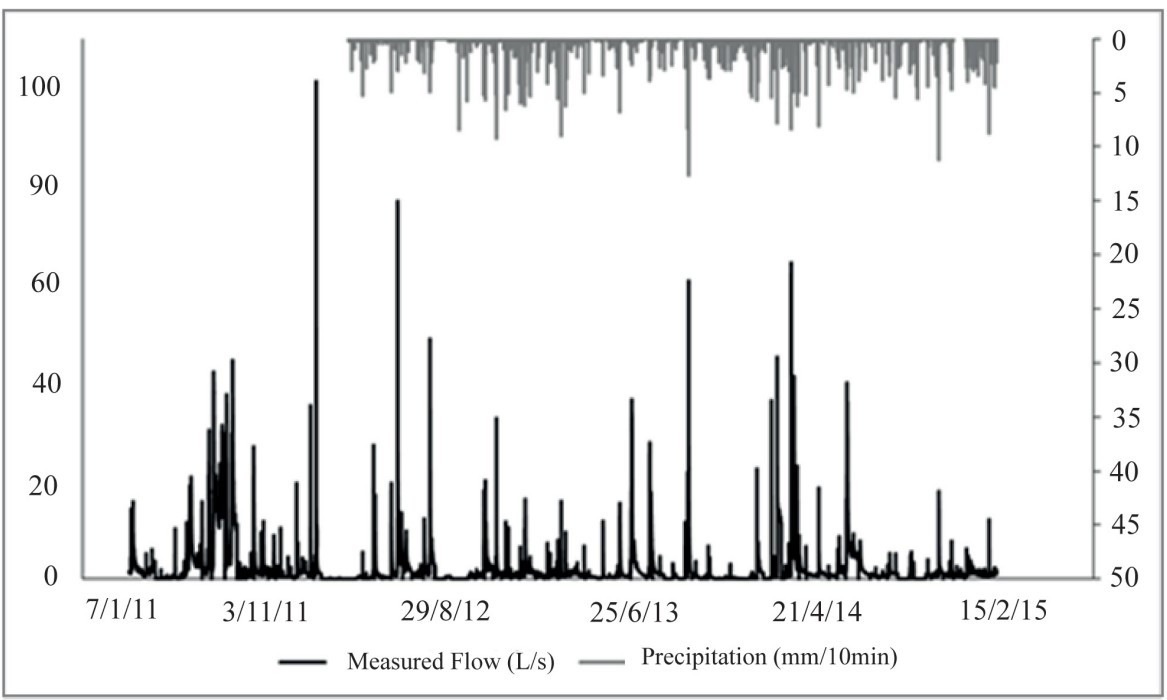

**Figure 2.** Measured flow and Precipitation data.

The soil water content was estimated by using the water retention curves and the water tension data measured at the depths of 0.10 m, 0.20 m, 0.30 m, 0.40 m, 0.60 m, 0.75 m, and 0.90 m. For determining the time interval (T) used for Tank Model simulations to a certain basin area A (km$^2$), (Sugawara, 1995) suggested the following equation:

$$T = 0.15\sqrt{A} \tag{1}$$

Since A = 0.0524 km$^2$ in the present study, T is 0.034 h, approximately 2.1 min. As the field measurement was conducted every 5 minutes, the value of 5 minutes was adopted for T.

The hydrological and the water tension data series presented many periods of no-data caused by technical problems with the sensors. For this reason, it was not possible to use the entire series of data obtained by the monitoring. Thus, there were selected 5 rainfall events for calibration and 2 for validation which all lasted 3 days. Table 2 shows the period, the mean, cumulative, measured and calculated values of precipitation and flow and the mean and cumulative values of evapotranspiration for the calibration events: I (occurred between January 14th to 16th, 2012), II (occurred between April 28th to 30th, 2012), III (occurred between June 4th to 6th, 2012), IV (occurred between June 20th to 22nd, 2013), V (occurred between March 8th to





**Table 2.** Hydrological data of rainfall events used in calibration and validation.

|  | I | II | III | IV | V | VI | VII |
|---|---|---|---|---|---|---|---|
| Mean precipitation (mm/5min) | 0.063 | 0.046 | 0.140 | 0.120 | 0.100 | 0.120 | 0.160 |
| Accumulated Precipitation (mm) | 54.90 | 40.60 | 122.92 | 107.13 | 84.62 | 104.59 | 139.12 |
| Average flow (m³/s) | 0.0037 | 0.0037 | 0.014 | 0.014 | 0.014 | 0.0044 | 0.011 |
| Average flow (mm/5min) | 0.021 | 0.021 | 0.080 | 0.080 | 0.080 | 0.025 | 0.060 |
| Accumulated flow (mm) | 17.82 | 18.33 | 71.81 | 68.86 | 64. 92 | 21.45 | 23.11 |
| Average ETR (mm/5min) | 0.00310 | 0.00150 | 0.00001 | 0.00300 | 0.00500 | 0.00150 | 0.00130 |
| Accumulated ETR (mm) | 2.72 | 1.32 | 0.07 | 2.34 | 4.26 | 1.32 | 1.14 |

10th , 2014); and validation events: VI (occurred between July 20th to 22nd, 2013) and VII (occurred between April 25th to 27th , 2012).

## 2.3 Tank Model

According to Sugawara (1995), the number of tanks used in the modeling depends on the basin size, the soil type and use, and the time interval used in the simulation. In the case of large basins, and long-term simulations with daily data, there is a strong influence of the base flow, so a 4-tanks structure is used to represent the hydrological processes. In the case of small basins, where short rainfall events will be evaluated, the greatest contribution to the flow will come from the surface runoff because the time response of the basin is too fast, and a few tanks are needed. For this reason, a 2-tanks structure was chosen for simulating the rainfall-discharge processes and estimating of soil moisture (Figure 3 (a)).

Then, the water balance equations were considered. Two versions of the Tank Model were applied in the study area. At first, the lumped Tank Model was created in order to obtain the parameters to be used in the distributed model. Secondly, apart from the parameters encountered in the first step, the distributed Tank (D-Tank) Model was established.

The application of the lumped model was carried out by considering the structure composed of two tanks (reservoirs), and the equation for the flow calculation was obtained as follows:

$$qs1 = a1 \cdot (S1 - HA1) \tag{2}$$

$$qs2 = a2 \cdot (S1 - HA2) \tag{3}$$

$$qs3 = b1 \cdot S2 \tag{4}$$





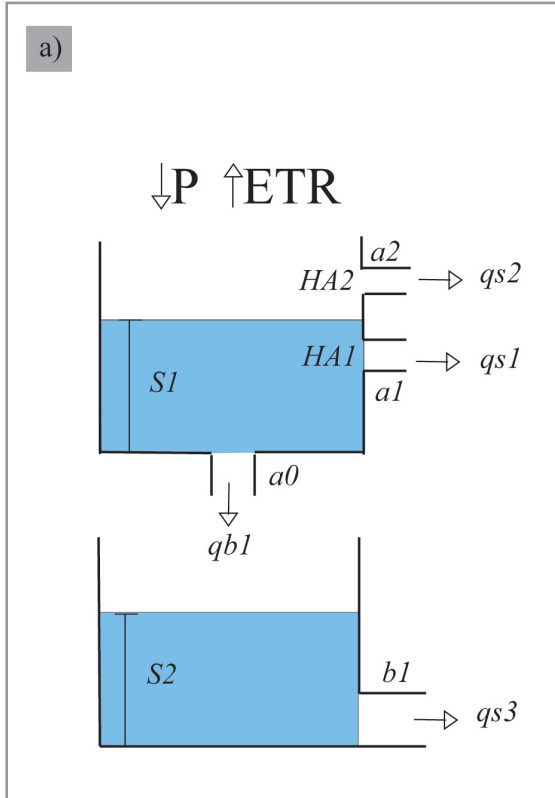
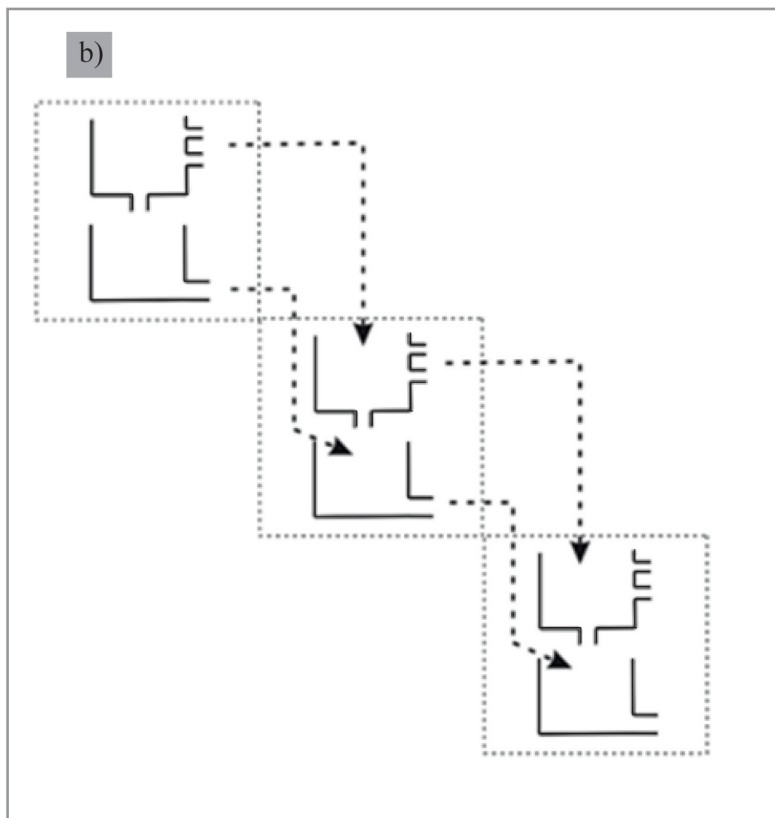

**Figure 3.** Structure of the Tank Model. (a) in a single cell; and (b) schematic representation of the movement of the flow between cells. P is the precipitation and ETR is the evapotranspiration, both in mm/5min.

125    $qb1 = a0 \cdot S1$            (5)

where S is the height of water stored in each tank (mm); qs2, qs1 and qs3 represent the surface, sub-surface and groundwater flow, respectively (mm/5min); qb1 is the percolation from the upper to the lower tank (mm/5min); a1, a2, and b1 are the coefficients of discharge; and a0 the coefficient of percolation; and HA1 and HA2 are the heights of the upper tank holes.

     The present study calls the flow and percolation coefficients of all the holes, and the heights of the lateral and vertical holes

130   of each tank as model parameters. These parameters´ values are usually obtained through calibration. According to Ishihara and Kobatake (1979), they are related to the soil type and use as well as to geological characteristics of the basin.





For the application of D-Tank Model, the study area was divided into square cells, with 2-m resolution. In each of them, the lumped Tank Model with the equations (2) to (5) was applied.

The flow traveled through the cells depending on slope, where the upper tank of one cell passed to the upper tank of the next cell (Figure 3 (b)). Thus, each of the tanks cell received flow generated by neighboring cells, which calculation is as follows:

$$Q = \Sigma c \cdot \Sigma j \cdot q_{i,j} \tag{6}$$

where Q is the total flow received by the adjacent tank; q is the flow calculated by lateral orifice; i represents the tank (1: upper tank, 2 lower tank); j is the lateral hole; and c is the number of cells that send outflow directly to the considered cell. The water storage in each tank was equal to that used by Kato et al. (2005), and determined by the following equations:

$$S1_t = S1_{t-1} + P - ETR + \frac{Q1}{Ct} - \Sigma j \cdot q_{1,j} \tag{7}$$

$$S2_t = S2_{t-1} + qb1 + \frac{Q2}{Ct} - \Sigma j \cdot q_{2,j} \tag{8}$$

where P is the precipitation; Ct is the total number of covered cells; Q1 and Q2 are the total flow received by the upper and lower tank, respectively; and t is the time step.

The present work adopted the D-infinity method proposed by (Tarboton, 1997) to determine flow direction between the neighboring cells. This method calculates the flow direction (represented by an angle between 0 and $2\pi$ rad), which is determined by the steepest direction of the eight triangular facets formed on a grid of 3x3 cells, centered on the pixel of interest.

## 2.4 Soil Water Index

As aforementioned, there are two indexes derived from the Tank Model which indicate the soil moisture condition in a basin: Soil Water Index – SWI (Okada et al., 2001), and the Tank Moisture Index – TMI (Lindner and Kobiyama, 2009). Both were analyzed for their applicability to prediction and understanding of different natural disasters, the first one focusing on landslides (Okada et al., 2001), and the second one on flood and drought occurrences (Lindner and Kobiyama, 2009). Because of its simplicity, the SWI was adopted in the present study.

According to Okada et al. (2001), the SWI is defined as the sum of the storage heights (S1, S2) of the Tank Model, which indicates the soil moisture condition of the basin in the lumped version and of each cell in the distributed version, i.e.,

$$SWI = S1 + S2 \tag{9}$$

## 2.5 Topographic Wetness Index and HAND

Beven and Kirkby (1979) proposed the Topographic Wetness Index (TWI) to quantify topographic control on hydrological processes. This index is calculated by

$$TWI = \ln \frac{\alpha}{\tan \beta} \tag{10}$$





where $\alpha$ is the flow accumulation in a catchment area; and $tan\beta$ is the slope. In the present study, the flow accumulation was calculated by using D-infinity algorithm. The slope was calculated in degrees, and after transformed into radians.

The Heigh Above the Nearest Drainage (HAND) proposed by Rennó et al. (2008) was calculated using the software TerraViewHidro. The generated HAND topology was reclassified aiming to better represent the basin characteristics.

## 2.6 Calibration and Uncertainty Analysis of Model Parameters

In order to perform an automatic calibration and uncertainty analysis of model parameters, the present work applied the automatic calibration algorithm Differential Evolution Adaptive Metropolis (DREAM(ZS)) proposed Laloy and Vrugt (2012) and Vrugt (2016). DREAM (ZS) is used for predict uncertainty in hydrological models (Cunha David et al., 2019), and is used for calibration of soil moisture distributed models (Linde and Vrugt, 2013). DREAM algorithm uses Bayesian inference for
estimation of model parameter values and their uncertainty. The used number (N) of Markov chains was 3 and the number (T) of generations was 15000.

In the present study a generalized likelihood function (GL) proposed by Schoups and Vrugt (2010) was used for the inference of the hydrological model parameters. The residual model used a homoscedastic Gaussian likelihood distribution.

The last 7500 sets of parameters sampled with the DREAM (ZS) algorithm were used to represent the uncertainty associated
with the parameter values and to create the probabilistic streamflow simulations. The present study evaluated the performance of the models through two functions: the Nash-Sutcliffe efficiency coefficient (NASH) proposed by Nash and Sutcliffe (1970) and the root mean square error (RMSE):

$$NASH = 1 - \frac{\sum_{n=1}^{T}(Qobs(t) - Qsim(t))^2}{\sum_{n=1}^{T}(Qobs(t) - \overline{Q}obs)^2} \tag{11}$$

$$RMSE = \sqrt{\frac{1}{t} \cdot \sum_{n=1}^{T}(Qobs(t) - Qsim(t))^2} \tag{12}$$

where Qobs(t) and Qsim(t) are the observed and simulated flows at the time t, respectively; and $(\overline{Qobs})$ is the mean value of the observed flow along the horizon t = 1 to T. The NASH values vary from $-\infty$ to 1. The RMSE is always positive, and RMSE = 0 means the perfect adjustment. These two functions were used also for evaluating the model validation.

## 2.7 Application Procedure

The first step of the work was the MATLAB implementation of the Tank Model and D-Tank Model which is similar to that
proposed by Kato et al. (2005). At this step, the algorithm was also implemented to calculate the flow direction in the study area.

After, when the distributed version was implemented, it was necessary to divide the study area into square cells with 2-m side, because the available DEM had 1-m resolution, and in the present study it was adopted 2-m resolution. This procedure was




**Table 3.** Range of the Tank Model parameters used in the calibration.

| Parameter | Minimum | Maximum |
|---|---|---|
| S1I (mm) | 0.0 | 1.5 |
| S2I (mm) | 70.0 | 75.0 |
| HA2 (mm) | 0 | 70 |
| HA1 (mm) | 0 | 35 |
| a1 ($5min^{-1}$) | 0.0001 | 0.09 |
| a2 ($5min^{-1}$) | 0.0001 | 0.09 |
| a0 ($5min^{-1}$) | 0.00001 | 0.009 |
| b1 ($5min^{-1}$) | 0.0001 | 0.09 |

performed in the ArcMap geoprocessing software, through the ETGeowizards extension which allowed the grid generation,
used to calculate the flow directions used in the D-Tank Model.

At the following step, 5 rainfall-events were selected for the calibration of the Tank Model through the DREAM(ZS) algorithm. For each event, a set of parameters was chosen, and this set was also applied to each cell of the D-Tank Model.

This transfer of parameters was validated as follows: for each parameter found in the five events of the previous step, its average was calculated to obtain a single set of parameters to be used for the model validation with two rainfall events.

As aforementioned, the parameters used in the validation step were those corresponding to the mean values of the parameters obtained in the calibration. Two periods were selected for validation of the distributed model, both with 3-days duration.

For the events of the validation period, the SWI values were compared through the linear correlation coefficient (r) with the calculated soil moisture values. Furthermore, the SWI values were also compared with TWI and HAND values.

## 3   Results and Discussion

### 3.1   Calibration and Uncertainty Analysis

The DREAM (ZS) was performed within a range pre-established by the user with the minimum and maximum limits of the decision variables which are the Tank parameters. Those limits are listed in Table 3.

Figure 4 presents the predictive uncertainty for each event selected for calibration, which permits to understand how the uncertainty in the model parameters translates into Tank Model uncertainty. The model seems to be able to match the peak
of the hydrographs, but the representation of the smaller flow values could be better. The events I, IV and V possess less uncertainty than events II and III. The set of parameters selected after the automatic calibration for each of the 5 analyzed events is listed in Table 4. These values were used later by the D-Tank Model.



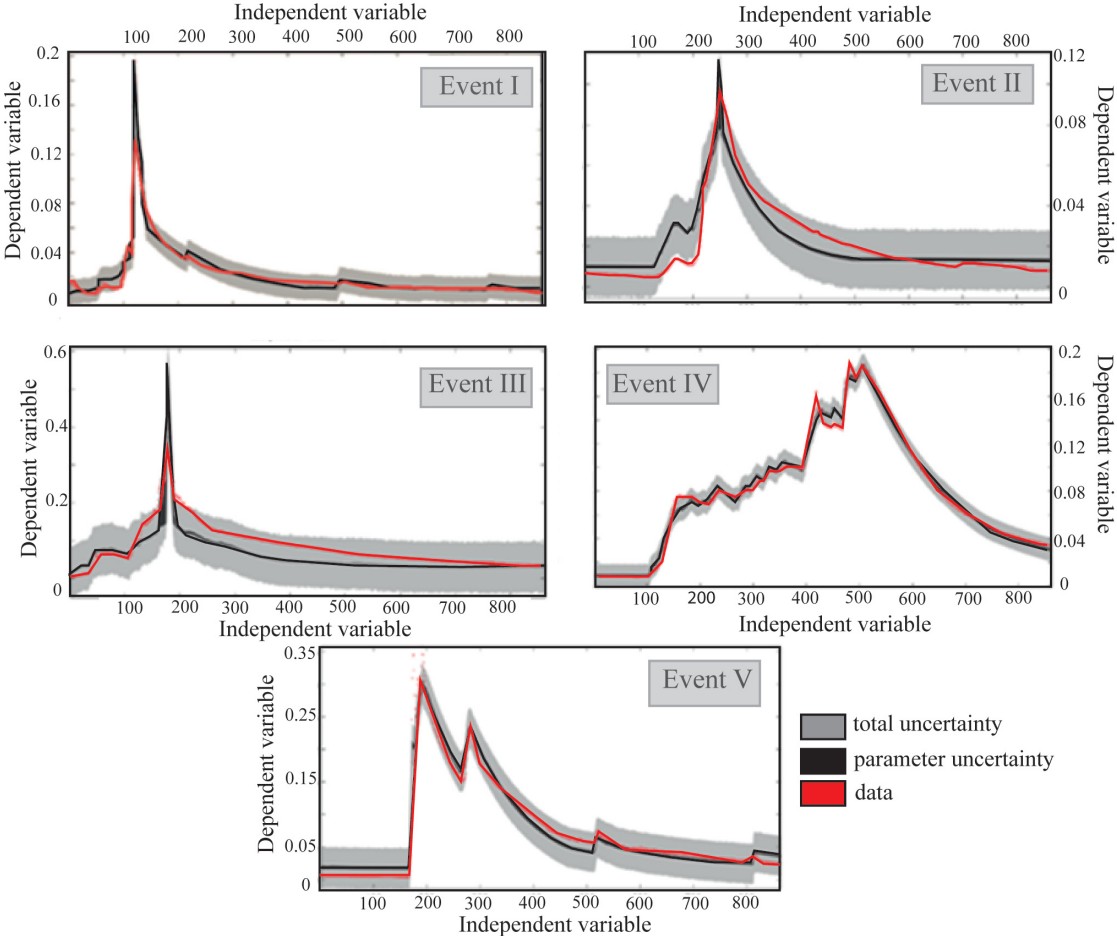

**Figure 4.** Observed runoff series (red), 95% uncertainty (light grey) and uncertainty associated with the values of the parameters (dark grey).

Table 5 shows the performance of the Tank Model and the D-Tank Model evaluated with two objective functions (NASH and RMSE) for each event. The values of both functions indicate a good fit of the Tank Model as well as the D-Tank Model, and that the set of parameters of the Tank Model is suitable for that of all the cells in the D-Tank Model. It is probably because the study area has uniform land-use and that precipitation and evapotranspiration were also considered uniform over the whole basin area.

In general, the lumped model presented a reliable performance in relation to the observed hydrograph, both in magnitude and in time, especially at the peaks. There was an underestimation event occurred during the period June 4th to 6th, 2012 (Event III), in which a peak flow was observed from 0.43 mm/5min, and other occurred between April 28th and 30th, 2013 (Event II),




**Table 4.** Tank Model parameters calculated in the calibration for each event.

| Parameter | Event I | Event II | Event III | Event IV | Event V | Mean |
|---|---|---|---|---|---|---|
| S1I (mm) | 0.20 | 1.28 | 1.16 | 1.50 | 1.50 | 1.045 |
| S2I (mm) | 72.87 | 74.61 | 72.87 | 75.00 | 75.00 | 73.68 |
| HA2 (mm) | 2.00 | 18.18 | 8.06 | 8.00 | 10.00 | 7.79 |
| HA1 (mm) | 28.40 | 55.32 | 35.00 | 50.00 | 50.00 | 49.94 |
| a1 ($5min^{-1}$) | 0.0020 | 0.0038 | 0.0043 | 0.0046 | 0.0056 | 0.0032 |
| a2 ($5min^{-1}$) | 0.0500 | 0.0329 | 0.0031 | 0.0003 | 0.0090 | 0.0347 |
| a0 ($5min^{-1}$) | 0.0069 | 0.0009 | 0.0057 | 0.0019 | 0.0020 | 0.0032 |
| b1 ($5min^{-1}$) | $9.42\cdot10^{-5}$ | $9.09\cdot10^{-5}$ | $3.63\cdot10^{-4}$ | $2.70\cdot10^{-5}$ | $2.70\cdot10^{-5}$ | $6.69\cdot10^{-5}$ |

**Table 5.** Calibration performance of the Tank Model (TM) and the D-Tank Model (DTM) evaluated with two objective functions for 5 events.

| | NASH | RMSE | NASH | RMSE |
|---|---|---|---|---|
| Event | TM | | DTM | |
| I | 0.94 | 0.005 | 0.72 | 0.011 |
| II | 0.91 | 0.006 | 0.72 | 0.010 |
| III | 0.92 | 0.015 | 0.85 | 0.022 |
| IV | 0.94 | 0.011 | 0.89 | 0.016 |
| V | 0.89 | 0.025 | 0.88 | 0.026 |

which had an observed peak flow of 0.095 mm/5min. In the other events, the maximum flows were close to 0.2 mm/5min, and the calibration was considered efficient in adjusting the hydrograph peaks.

In relation to the lower flows, the lumped model corresponded adequately to three events (I, II and III), and in the others, it underestimated the minimum flows.

The calculated flow by the distributed version also showed good correspondence, maintaining the good peaks adjustment. There was a tendency to overestimate the flows of intermediate values (Figure 5).

The NASH values obtained by the D-Tank Model varied between 0.72 and 0.89, and the RMSE from 0.01 to 0.026. Presenting a slight decrease in performance over the lumped model, the D-Tank Model still showed a satisfactory adjustment to the observed data.





**Figure 5.** Calibration results with Tank Model and D-Tank Model.

## 3.2 Validation

The Tank Model simulation resulted in a satisfactory water balance for the two events analyzed, the RMSE close to zero and NASH very close to 1 (Table 6).

Figure 6 shows the predictive uncertainty of the events selected for validation. The uncertainty in validation increased in relation to the events selected for calibration.





**Table 6.** Calibration performance of the Tank Model (TM) and the D-Tank Model (DTM) evaluated with two objective functions for 5 events.

| | NASH | RMSE | NASH | RMSE |
|---|---|---|---|---|
| Event | TM | | DTM | |
| VI | 0.96 | 0.006 | 0.95 | 0.007 |
| VII | 0.92 | 0.010 | 0.72 | 0.020 |

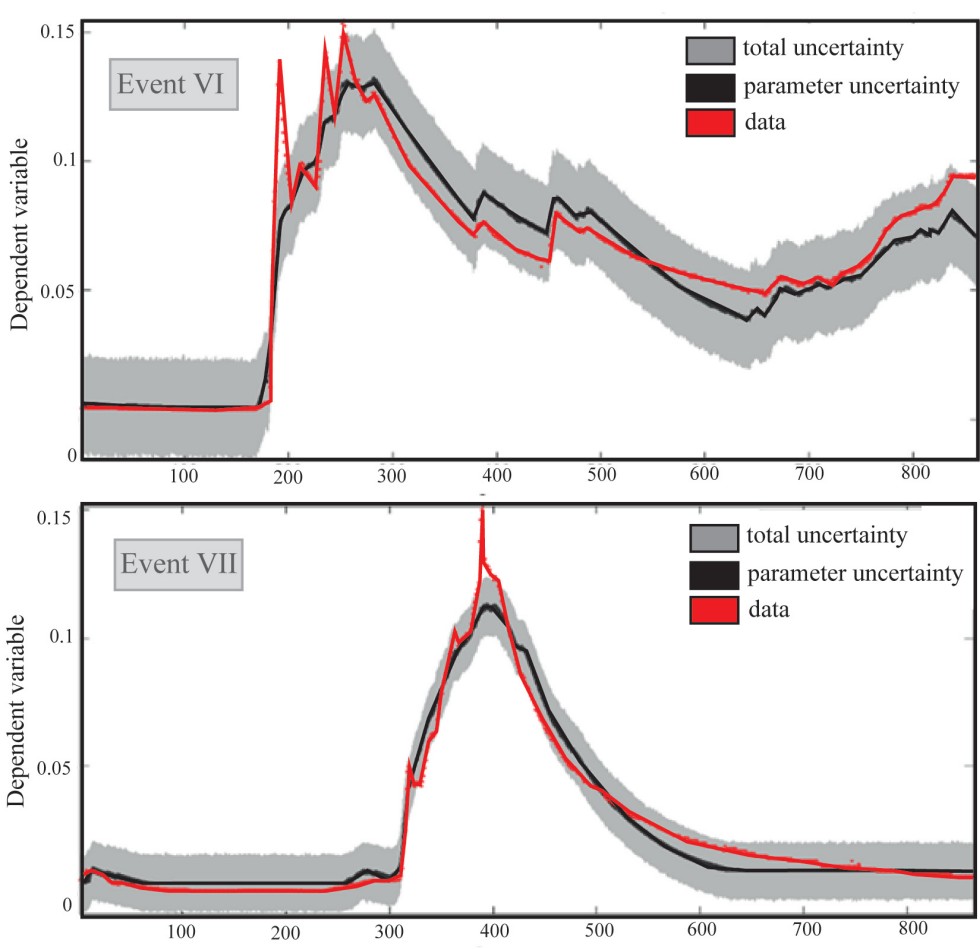

**Figure 6.** Uncertainty analysis for validation. Observed runoff series (red), 95% uncertainty (light grey) and uncertainty associated with the values of the parameters (dark grey).





**Figure 7.** Validation results with Tank Model and D-Tank Model.

In the similar way, the D-Tank Model showed a good performance for the event VI, with RMSE also close to zero, and NASH
close to 1. In the event VII, there was a slightly high overestimation. The hydrographs representing the validation simulations
are shown in Figure 7. The two versions (lumped and distributed) of Tank Model overestimated the flows after the peak in
the event VII. It can be attributed to the used parameters, especially the infiltration coefficient and the baseflow coefficient,
which could be probably overestimated. In the event VI, the Tank Model underestimated the maximum flow peak, which was

enhanced by the D-Tank Model.



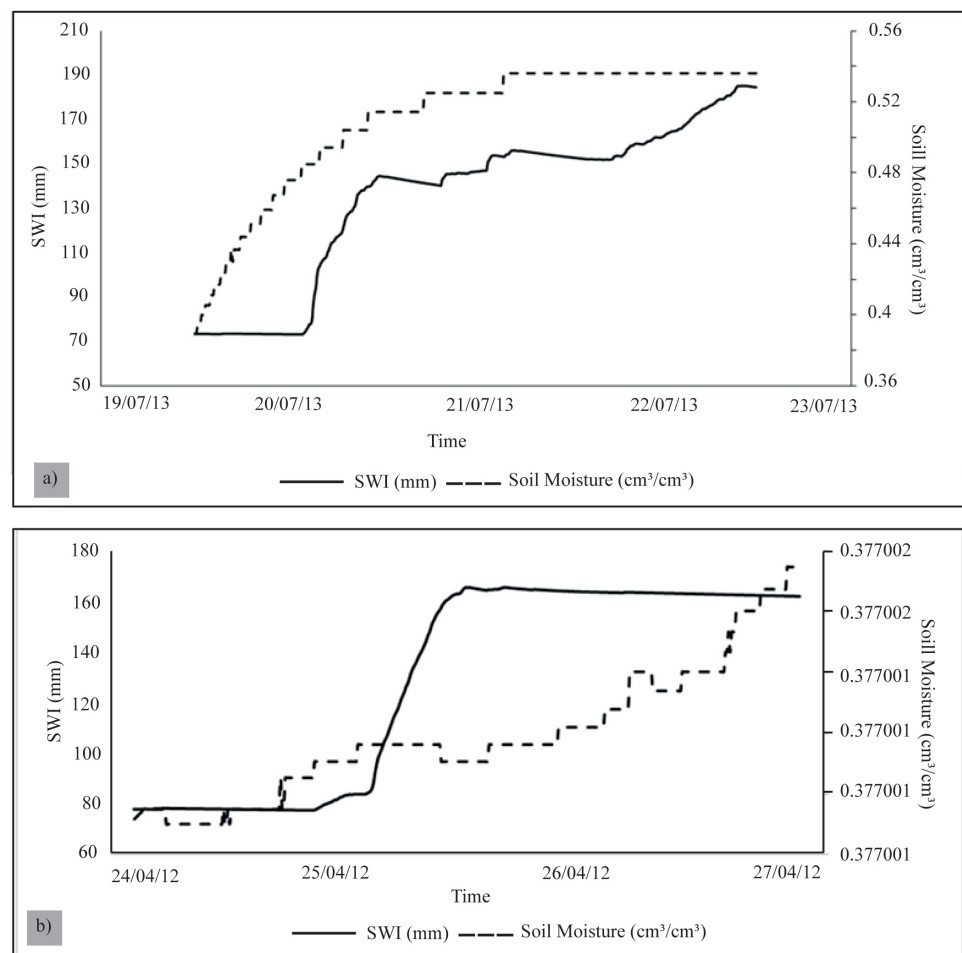

**Figure 8.** Relation between SWI and soil moisture calculated by the values obtained at the tensiometer: (a) A3 location at 10-cm depth during the Event VI; and (b) Z4 location at 30-cm depth, during the Event VII.

These results show that the distributed model was adequate for the application of the following steps (SWI generation and point comparison with the soil moisture values estimated from the tensiometer data).

### 3.3 Soil Water Index

Figure 8 shows two examples of behavior comparison between the SWI values obtained with the D-Tank Model simulation
and the soil moisture values calculated by (a) the tensiometer A3 (10 cm depth) for the period July 20th to 22nd, 2013 (event VI) and (b), the tensiometer Z4 (30cm depth) for the period April 25 to 27th, 2012 (event VII). A visual analysis of the Figure 8 implies a satisfactory performance of the SWI gained from the D-Tank Model.



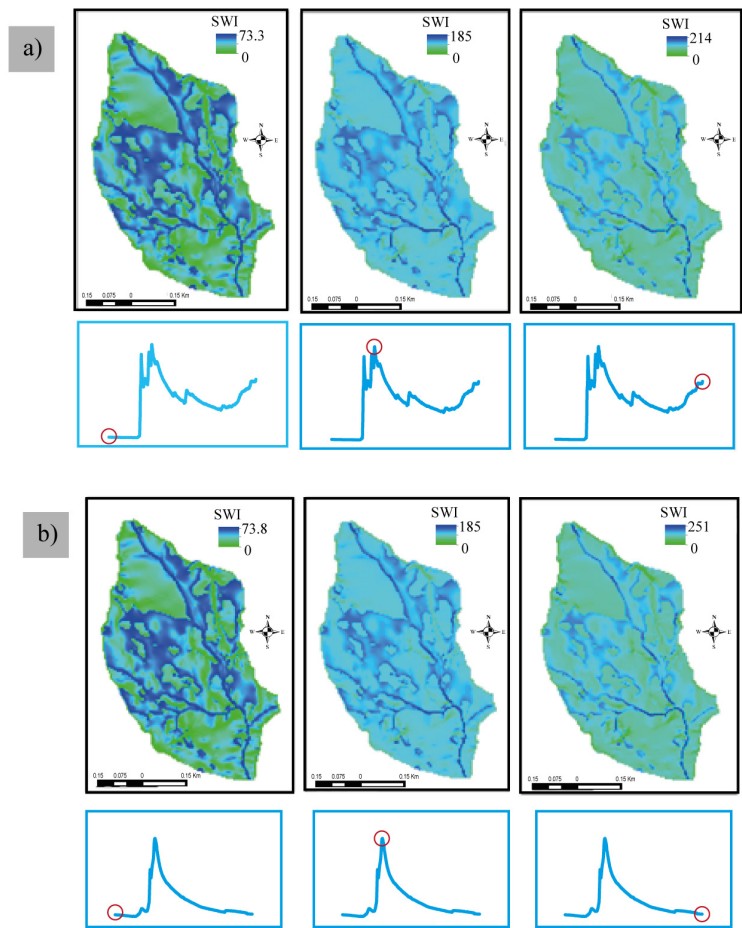

**Figure 9.** Spatial variation of the SWI values (mm) at 3 different moments of a hydrograph: (a) event VI; and (b) event VII.

The spatial variation of the SWI values at three different moments (the beginning of the event, the peak, and the last moment) of the two validation events VI and VII were elaborated in Figure 9. In both events the SWI behavior was very similar, especially

at the beginning. It resulted from the fact that the initial storage heights of the two reservoirs were the same in both events. In the SWI maps related to the peak of the hydrograph, there is also a similarity between the values in the two events, which is consistent, since both had the same maximum flow rate in 0.15 mm/5min. In general, the points of the highest SWI values were located along the drainage network. However, high SWI values were also observed at certain points of hillslopes. Furthermore, the SWI values were larger in the area closer to the basin exit, which is reasonable.





At the end point of the hydrograph, there was a difference in the SWI values. In the event VI, the maximum SWI value was higher, reaching 251 mm, while most of the basin was showing its values of up to 193 mm. In the event VII, the maximum value was slightly smaller (214 mm), however the remaining area of the basin was wetter, represented by a SWI of 205 mm.

    Between the two events, the spatial behavior of the SWI was similar, where wetter areas at any time kept their wetness under other circumstances (Figure 8). In relation to the upstream section, the SWI values in the section closer to the outlet were
slightly larger, which is coherent.

    Vachaud et al. (1985) observed this behavior of soil moisture and proposed the concept of temporal stability of soil moisture, i.e., there is a high probability that a moist condition for a moment will remain itself at other times. By using geostatistical tools, Gonçalves et al. (1999) observed the persistence in time of moisture distributions. This is an interesting result because the confirmation of the temporal stability of the soil moisture distribution allows reducing the sampling number for monitoring
or estimating the soil water storage.

    The spatial variation of the SWI can be observed also in Figure 10, where two cross sections were selected for SWI evaluation at the peak of the event VI. In the cross-section 1, there is a 60-m elevation difference between the highest point and the river, although these points are about 200-m distant.

    The SWI value in this profile was almost constant, with a slight increase of 20 mm in the point located in the river. In the
cross section 2, the SWI behavior in a profile closer to the outlet is presented. With a difference of 30 m of elevation between the highest point and the river, but distant from one another only by 80 m, the SWI variation throughout the section was slightly larger, and, as expected, the highest SWI value occurred at the river.

### 3.4   Linear Correlation between SWI and Soil Moisture

For linear correlation analysis between SWI and the only measured values of soil moisture, the present study used only the
tensiometers' data which were considered minimally coherent in relation to the precipitation of the event. Therefore, the tensiometer of cell A12 was excluded from all the comparisons.

    In the event VI, a strong correlation between the calculated SWI and the soil moisture in all the analyzed cells and depths. They are all below 0.7 (Table 7). Figure 11 shows the linear relation between SWI and Soil Moisture calculated by the tension of to the sensor A3, tensiometer location at 10 cm of depth during the Event VI.

Significant correlations between the SWI and the soil moisture were also obtained for the event VII (Table 8). However, in this event, those correlations were more significant in the uppermost layers of the soil. It can be said that all the analyses demonstrated that a SWI behavior is similar to that of the soil moisture.

    TWI and HAND were generated (Figure 12) to be used as soil moisture indicators. Both maps were compared to SWI results (Figure 9) in order to verify a correlation between them. A visual analysis permits to verify the similarity of spatial behavior
between TWI and SWI maps. The HAND topology also shows the similarity with TWI (the lower values are located in the areas more close to the drainage, that are usually more moist).





**Figure 10.** Variation of SWI in two transverse sections of the basin at the hydrograph peak for event VI.

The results of the correlation analysis show the high correlation of SWI with the TWI values (in both events and in all 3 different hydrograph moments) (Table 9). HAND presented smaller values of r, but still presented significant correlation with SWI.

The correlation between SWI and TWI could be explained for the structure of the D-Tank Model, where similarly as the TWI, the accumulated flow has a determinant influence on the generated results. Thus, it can be said that SWI has a major relation with flow accumulation and slope than the vertical distance to the stream.





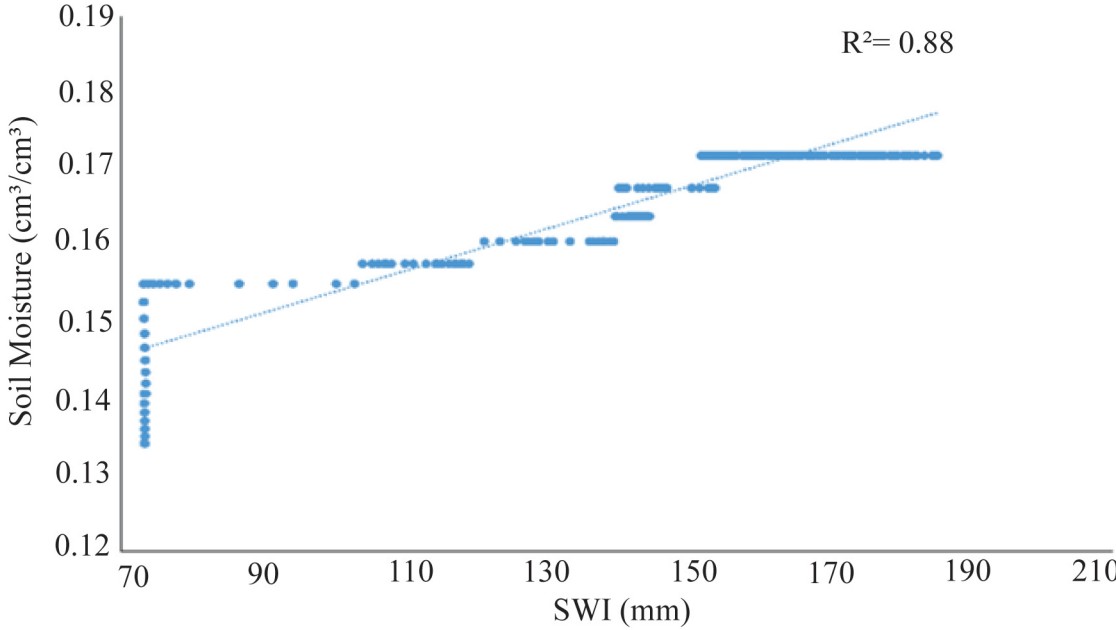

**Figure 11.** Linear relation between SWI and Soil Moisture calculated with the values obtained at the tensiometer A3 location at 10cm of depth during the Event VI.

### 3.5 Linear Correlation of Soil-Moisture Spatial Distribution with SWI, TWI and HAND

For spatial analysis of linear correlation between the Soil Moisture and SWI, TWI and HAND, the present study used only
the soil moisture data estimated with the tensiometers located at 30-cm depth, because of the two reasons: (i) these were the sensors that generated more coherent results at all the analyzed events; and (ii) Ehret (2014) reported that the mean moisture of the first 30-cm soil layer was a robust indicator of the slopes initial conditions prior to a rainfall event.

Several studies used geostatistics for estimate soil moisture patterns from point observations (Bardossy and Lehmann,1998; Western et al.,1998; Western and Bloschl,1999; Western et al.,1999; Perry and Niemann,2008; Yang et al.,2017). The Inverse
Distance Weighting (IDW) method was used for interpolating the soil moisture values calculated by tensiometer information, and the distributed soil moisture maps were generated. However, in this case, only the covered area by the sensors was treated because of the limitation of this method (Figure 13).

The different patterns of spatial distribution of soil moisture can be observed between two analyzed events (Event VI and Event VII). Though the total values of rainfall of the Event VI and VII were similar (104.59 mm and 139.12 mm, respectively),
the values of the initial soil water content were slightly different (0.55 cm$^3$/cm$^3$ in Event VI and 0.41 cm$^3$/cm$^3$ in Event





**Table 7.** Values of r for the comparison between SWI and soil moisture for Event VI

| | Sensor Depth (m) - Configuration I | | | | | | |
|---|---|---|---|---|---|---|---|
| | 0.1 | 0.2 | 0.3 | 0.4 | 0.6 | 0.7 | 0.9 |
| A4 | - | 0.80 | 0.80 | 0.90 | 0.90 | - | - |
| A3 | 0.88 | - | 0.60 | 0.83 | 0.65 | - | 0.65 |
| B3 | - | 0.60 | 0.70 | 0.70 | 0.80 | 0.80 | 0.90 |
| B4 | - | 0.80 | 0.90 | - | - | 0.90 | 0.90 |
| Z3 | - | 0.70 | 0.70 | - | - | 0.83 | - |
| Z4 | - | - | 0.70 | 0.80 | 0.90 | 0.90 | 0.80 |
| | Sensor Depth (m) - Configuration II | | | | | |
| | 0.1 | 0.3 | 0.4 | 0.1* | 0.3* | 0.4* |
| B12 | - | 0.90 | 0.80 | 0.80 | 0.90 | 0.90 |
| Z12 | - | 0.90 | 0.90 | - | 0.90 | 0.90 |

**Table 8.** Values of r for the comparison between SWI and soil moisture for Event VII

| | Sensor Depth (m) - Configuration I | | | | | | |
|---|---|---|---|---|---|---|---|
| | 0.1 | 0.2 | 0.3 | 0.4 | 0.6 | 0.7 | 0.9 |
| A4 | 0.60 | 0.90 | 0.90 | 0.30 | - | 0.30 | - |
| A3 | 0.80 | 0.90 | 0.70 | 0.90 | 0.40 | 0.40 | |
| B3 | 0.60 | 0.60 | 0.70 | - | - | - | - |
| B4 | 0.80 | - | 0.80 | 0.80 | - | 0.40 | - |
| Z3 | 0.30 | - | 0.90 | - | - | - | 0.9 |
| Z4 | 0.50 | 0.90 | 0.90 | 0.80 | - | 0.70 | - |
| | Sensor Depth (m) - Configuration II | | | | | |
| | 0.1 | 0.3 | 0.4 | 0.1* | 0.3* | 0.4* |
| B12 | 0.60 | 0.95 | - | 0.60 | - | - |
| Z12 | - | 0.60 | - | 0.80 | - | 0.90 |

VII), which might result in the differences in the peak flow between these events (0.15mm in Event VI and 0.09mm in Event VII). Zehe et al. (2007) and Uber et al. (2018) found significant effects of initial soil moisture condition on discharge values.





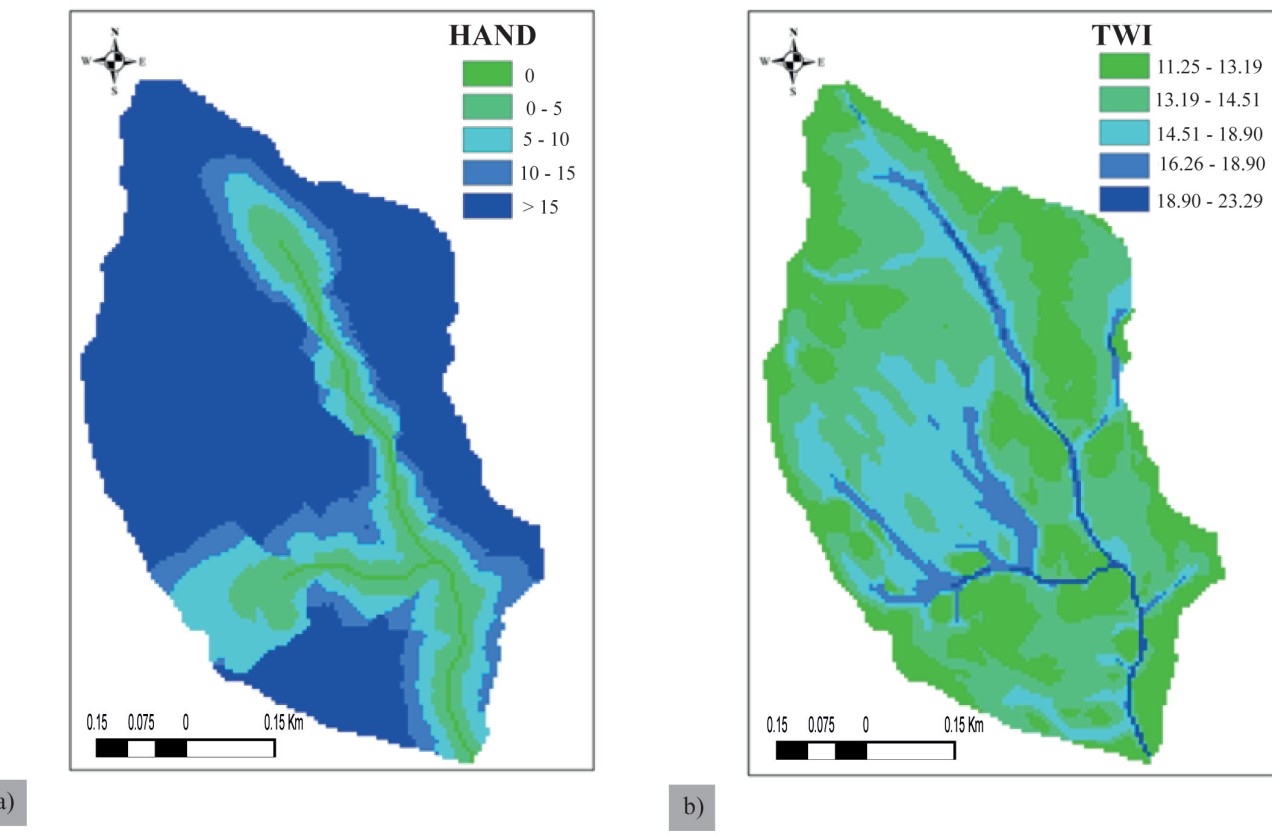

**Figure 12.** Spatial distribution in the study basin: (a) HAND topology; and (b) TWI.

**Table 9.** Values of r for the comparison between the SWI, TWI and HAND

|  | Event VI | | | Event VII | | |
|---|---|---|---|---|---|---|
|  | I | II | III | I | II | III |
| TWI | 0.88 | 0.88 | 0.88 | 0.88 | 0.88 | 0.87 |
| HAND | 0.62 | 0.63 | 0.62 | 0.62 | 0.63 | 0.62 |

Furthermore, Zehe and Sivapalan (2009) demonstrated how the initial soil moisture condition influenced on the infiltration process and consequent runoff generation.

The distribution features of the soil moisture had high correlations with SWI, TWI, and HAND (Table 10).





**Figure 13.** Soil Moisture (cm$^3$/cm$^3$) at 30-cm depth, calculated with IDW method, for three different stages of the events VI and VII.

**Table 10.** Values of r for the comparison of the distributed Soil Moisture with SWI, TWI and HAND

|  | Event VI | | | Event VII | | |
|---|---|---|---|---|---|---|
|  | I | II | III | I | II | III |
| SWI | 0.95 | 0.95 | 0.95 | 0.95 | 0.95 | 0.95 |
| TWI | 0.90 | 0.90 | 0.90 | 0.90 | 0.90 | 0.90 |
| HAND | 0.84 | 0.84 | 0.84 | 0.84 | 0.84 | 0.84 |

According to Western et al. (1999), soil moisture patterns depend on different topographic properties at different times.The authors found that TWI can explain soil moisture variability at catchment scale. Minet et al. (2011) analyzed soil moisture





patterns in terms of runoff response, and concluded that the use of TWI for evaluating the soil moisture pattern is a suitable technique. Observing Table 10 more in detail, it can be said that the SWI generated by D-Tank Model performed slightly better than TWI. Hence, the use of SWI could be more recommended for the soil moisture representation. Loritz et al. (2019) discussed hydrologic similarity in different catchments comparing TWI and HAND performance, and concluded that despite the similarities between these indices, they represent different hydrological aspects, where the modified-HAND version performed better than TWI and HAND approaches. Contrary to their study, the present study showed that HAND was the worst index to represent soil moisture patters.

Downer and Ogden (2003) also used a distributed conceptual model to estimate soil moisture, concluding that physically-based hydrologic models can be used to make predictions of peak discharge and soil moisture. The present study can infer that a distributed conceptual hydrological model can be used to predict soil moisture patterns in catchment scale.

## 4  Conclusions

By using the D-infinity method of (Tarboton, 1997), the present study constructed a distributed version of Tank Model and
called it D-Tank Model. Adopting the SWI proposed by Okada et al. (2001), the D-Tank Model was applied in order to evaluate the soil moisture distribution in the Araponga river basin in the rural area of Rio Negrinho municipality of the Santa Catarina state, southern Brazil. Then the performance of the D-Tank Model and the spatially-distributed SWI values were verified with rainfall, discharge and soil moisture data.

Based on the obtained results, it can be concluded that the D-Tank Model had satisfactory performance for the studied
period. Hydrographs between the observed and calculated flow showed a good fitting. The spatially distributed values of the SWI, generated from the D-Tank Model, performed also satisfactorily, representing well the spatial variation of the water storage within the basin, which were qualitatively confirmed with the soil moisture data. Even compared with TWI and HAND, the SWI values presented better fit with the values of linear correlation between the analyzed maps.

The spatially-distribution analysis of the soil moisture was used to validate SWI, which shows good correlation. Thus, the
use of SWI could be recommended for the soil moisture representation. However, the D-Tank Model and its derived SWI values should be checked with other basins which are different from the Araponga basin in terms of the basin size, forest type, land-use, among others.

*Code and data availability.*  The full analysis scripts are published on Github (https://github.com/mvsofia/dtankmodel). The data is available upon request.

*Author contributions.*  The methodology was developed by SM, and supervised by MK. The data was provided by AM. All code was developed by SM. The manuscript was written by SM, with contributions of MK in the Introduction and Discussions, and AM in the Introduction.



AM supplied the field descriptions, and one of the figures. Structure and language of the manuscript was revised and improved by AM and MK.

*Competing interests.* The authors declare to have no competing interests



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
