# Peer review of "Evaluation of Soil Water Index of distributed Tank Model in a small basin with field data"

_Hydrology and Earth System Sciences, 2019_

## Referee Comment (RC1) · Anonymous Referee #1 · 24 Feb 2020

Review of **Evaluation of Soil Water Index of distributed Tank Model in a small basin with field data** by Vasconcellos et al.

The study of Vasconcellos et al. applied the Tank Model and its distributed version D-Tank to the Araponga river basin in Brazil. Here, soil moisture values were measured and the authors calculated the soil water index (SWI) of D-Tank. The authors correlated the soil moisture measurements with SWI. Afterwards, the SWI was correlated with the Topographical Wetness Index and Heigh Above Nearest Drainage. The authors conclude that SWI is a useful metric to represent soil moisture.

The study seems interesting, but I think some effort is needed to improve the manuscript. Generally, there are often sentences that seem off to me (see also my long list of minor comments), and I think the authors should go over it once more to capture all these small errors. Also in the Figures and Tables there are many minor issues as missing units or different color scales.

In addition, the authors make often statements which are hard to find back in the figures or need a bit more support. For example, based on Figure 8, the authors state that there is a satisfactory performance, but Figure 8b seems quite off in my view, so I am not sure if you can make this statement. More importantly, there are many more tensiometers and events, so why not show more (ideally even all) data? This would generalize and support the statements of the authors much more. The same applies to Figure 11, even though the correlation coefficients are reported in Tables 7 and 8, more results can easily be shown. In addition, a more critical look at this figure is probably justified, instead of just looking at the R-squared values. For example, why are the soil moisture values flattened off? And why is the SWI just changing after a soil moisture value of around 0.155 cm3/cm3? Just looking at the R-squared value gives a good indication of the linear relation, but it does not say much about, for example, a consistent bias, so I might be good to look a bit further here.
Related to this, I also think it is important that the authors assess the spatial correlation between SWI, HAND and TWI a bit more thoroughly, instead of just a linear regression. For example, normalizing them and subtracting the maps from each other will lead to insights where the values match, and where deviations start to occur. This can also easily be done for the interpolated values of soil moisture (Figure 13).

Regarding the calibration, what is actually the rational behind using the DREAM algorithm? First, I am a bit confused on how it was implemented, there is a mention of a generalized likelihood function, but in the next sentence (P9.L174) it is stated that the last 7500 samples are used to represent uncertainty. So how was the uncertainty actually represented in the end? In addition, the DREAM method is applied to each event to get parameter estimates, but in a next step, the parameter values of the different events are just averaged, which seems a bit simplistic in contrast to the DREAM algorithm. Why not give more weight to the parameters that are more likely? The added value of this method is also that it gives you uncertainty values, but afterwards, the authors do not really do anything with it in the analyzes of soil moisture indexes. So what is the point of using this method? In addition, I think there is also a strong influence of the chosen evaluation statistics, as the Nash-Sutcliffe efficiency and the RMSE have a strong bias for large values. In other words, in the event based approach in this study, when the height of the peak matches, high values for these metrics are likely to be found.
The authors also never specifically mention how they deal with the initial states, which will have a strong influence on the results. Is it correct that these are calibrated?

Lastly, I also wonder if it is really a surprise that TWI, HAND and SWI show good correlations. In the end, they are all based on the flow direction map, and especially on the event time-scale for a small

basin, I think they should show similar patterns. I am not sure about this, but I just wonder what the authors thoughts are here. Would it not be much more interesting to see if these findings still hold for a longer time-scale and/or even a larger area?

Concluding, I believe there are some substantial improvements needed. I hope the authors find my comments useful and I look forward to an improved manuscript.

**Minor comments**
P1.L18. Condition → conditions?
P1.L24. (Beven and Kirkby, 1979) → (Beven and Kirkby, 1979).
P2.25. Have been → has been
P2.L27. I do not think you can see the hand as model, more a corrected elevation map
P2.L65. clayey texture. → clayey texture
P2L67. Clay) → clay
P3.L75. Sentence seems odd, please rephrase.
P3.L78. At the 0.3m → at 0.3m
P3.L85. As showed → as shown
P5.L95. (Sugawara, 1995) → Sugawara, (1995)
P5.L100-101. there were selected 5 rainfall events → there were 5 rainfall events selected
P5.L111. I think this depends on the basin. Surface runoff will only occur if there is infiltration excess (so depending on the rainfall rate and infiltration capacity) or saturation excess (depending on the saturated state of the catchment).
P6.L120. I would suggest to use subscripts, it is not clear if it is H*A1 or HA1
P7.L131. soil type and use as well as → soil type as well as
P9.L162. And after transformed → and afterwards transformed
P9.L163. Please be more specific. HAND requires defining also the stream, as you have to define where the stream starts (i.e. number of upstream cells). Or do you have this mapped? And what do you mean with reclassified?
P9.L169. soil moisture distributed models → distributed soil moisture models?
P9.L167. proposed Laloy .. → proposed by Laloy …
P9.L168. Used for → used to
P9.L188. it was adopted 2-m resolution → a 2-m resolution was adopted
P10.L194. Why do you just use the average if you do such a complex calibration?
P10.L197. Should you also not look at the spatial distribution?
P13.L225. You cannot state you have a good water balance based on the NASH and RMSE values.
P15.L230. In the similar way → in a similar way?
P15.L233. Yes, the baseflow coefficient must be off, but why should this be overestimated? Also in calibration the declining limbs are off.
P16.L239. behavior comparison?
P16.L241. of the Figure → of Figure
P16.L242. How is it satisfactory? The lines in Fig. 8B are completely off.
P17.L249. I find this a bit hard to see, it is not really different compared to other values along the stream.
P18.L251-253. The legends actually say the opposite, that the maximum values for event VI is 251mm and for VII 214mm.
P18.L255. I cannot see from the figures that the values are higher at the outlet.
P18.L264. I cannot see this, can you add the location of the river in the figure?
P18.L269. the only measured → the measured
P18.L270. What do you mean with minimally coherent?

P18.L272. Sentence misses a verb.
P18.L273. What is the value of 0.7? Correlation coefficient?
P18.L274. of to the → of
P18.L275. Please also report significance levels and p-values.
P18.L280. The comparison would be easier if 12 a and b both use a continues, similar colorscale. You could also normalize the values for comparison.
P19.L282. How was this correlation analysis carried out? Does the regression use all the cell-based values of SWI and HAND? Why not show a graph of this?
P19.L285. Explained for → explained by
P19.L286. Major → stronger?
P20.L290. What do you mean with coherent?
P20.L293. for estimate → to estimate
P21.L296. How do you define the covered area?

Table 3. Are S1I and S2I the initial states? Do you calibrate these?
Table 5. Please add the units of the RMSE.  What is the range of values for NASH and RMSE here, based on the uncertainty ranges?
Table 7-8. Why are there so many places left empty? What are the criteria to leave these out?
Table 10. All the values are the same for the different events, is that correct? Are I, II, and III referring to the different moments in the hydrograph?

Fig1. Please add units of elevation.
Fig4. Please add the units. I also assume the dependent variable is flow, and the independent variable is time, correct? Can you add this instead? What do you mean with uncertainty of the parameters? How is this different from the grey area?
Fig5. What is the difference with Fig.4? What happened to the uncertainty margins?
Fig9. Please make sure all figures use the same color scale. It has hard to compare like this.
Fig10. The second graphs is also labeled as Cross Section I.
Fig11. Why are the soil moisture values flattened off? How is this regression carried out?
Fig13. Please use the same color scale for comparison.

---

## Referee Comment (RC2) · Nicolas Rodriguez (Referee) · 8 May 2020

This study by Vasconcellos et al. applies a distributed version of the "Tank Model" (DTM) to a relatively small catchment in Brazil in order to simulate soil water storage, derive a soil moisture proxy (Soil Water Index, SWI), and compare SWI to measured soil moisture and to topography-derived indexes (HAND, TWI) that could also be used as proxies of soil moisture. The goal of this approach is to show that a simple hydrological model can be applied in a distributed manner to estimate the spatial distribution of soil moisture in a catchment, which acts as a major control on hydrological and biogeochemical processes.

I think that there is an interesting idea at the origin of this work. However, in my opinion this manuscript would need considerable improvements to reach the quality required for publication in HESS, both in substance and format. I think that the methods could be carefully reconsidered and revised. At least, there could be clarifications and explanations about the choices that were made in the methods (especially on calibration and validation), and an extended discussion about the impact of these choices on the results and interpretations. The quality of the presentation of the results could easily be improved, in particular the equations (e.g. correct indexing and naming of symbols) and the figures (labels, axis names, legends…). Here are my major comments:

1. In this work it is not immediately clear why one should bother with a hydrological model which largely simplifies transport processes in the soils and comes with many sources of uncertainties (P, Q, ET, parameters) instead of using simple topographically-derived or precipitation-derived proxies of soil moisture. Why use a model if simpler metrics are satisfactory (especially when keeping table 10 in mind)?

2. Given the relatively small size of the catchment (6 ha), the assumption of spatial homogeneity of inputs / land use / soil water parameters, wouldn't a physically-based distributed model (e.g. CATFLOW, Zehe et al., 2001) be more appropriate to simulate soil moisture? This question arises specifically after such models are mentioned in the introduction (L30-36) but not discussed further. In addition, physically-based distributed models would allow a direct and meaningful comparison of simulated vs observed soil moisture.

3. The equations could be re-written in order to avoid current ambiguities:

- Using subscripts and superscripts for the indexes (0, 1, 2…)
- Writing any explicit dependence on time t with (t), and avoiding using a subscript for this
- Replacing t-1 by t-$\Delta t$
- Correctly placing the bound variables (i, j, c) below the sums and giving the lower and upper bounds
- Avoiding using a dot symbol between bound variables and elements in the sum (this will be confused with a product)
- Correcting equations 7 and 8; $\Delta t$ is missing to go from fluxes (L/T or $L^3$/T) to water amounts (L or $L^3$)

Most of my concerns are about the calibration-validation choices:

4. Would it not make much more sense to calibrate the DTM rather than transferring the parameters of the calibrated TM to DTM? In the end, the procedure would be the same, but the DTM would be used in the DREAM algorithm instead of TM. Seeing TM working better than DTM in both calibration and validation (tables 5 & 6) suggests that it should have been done

this way. As a result, the derived uncertainties are all coming from the calibration of TM, not from DTM which is the model actually used to derive SWI. Thus, the true uncertainties which will affect SWI estimations are not shown.

5. The models are calibrated to hydrographs and not soil moisture which is the target. Very good model fits to hydrographs may not guarantee very good fits to soil moisture (e.g. in a catchment dominated by groundwater responses, Loritz et al., 2017; Rodriguez et al., 2019). Is it not possible to similarly calibrate and validate DTM, but to soil moisture observations instead?

6. The visual comparisons for soil moisture are misleading. Why show only 2 locations out of 9? Is it really meaningful to compare SWI and soil moisture directly? Figure 8 suggests that their relative variations differ by orders of magnitude (especially in Fig8b), and that they may be poorly related (especially in Fig11, the step-wise behavior). The use of a correlation coefficient does not take additive and multiplicative biases into account (Legates and McCabe, 1999), so I suggest to use another performance measure. Also, the correlation coefficient does not allow a spatially-distributed comparison of the results with the observations, while it seems important for the aims of this study (see useful suggestion from first reviewer)

7. How were parameter sets chosen (L192)? I think that the use of an average parameter from all calibration events is not a standard method. In addition, no verification was done in order to check that the mean parameter set actually works well for the calibration events. In my opinion, the standard method would be to calibrate the DTM to all 5 events simultaneously, and validate to the 2 last events simultaneously. Shouldn't DTM work well for all events, to be representative of transport processes in the soils?

8. Was the Global Likelihood of Schoups and Vrugt really used? Why choose a constant homoscedastic gaussian likelihood error model, which is already available in DREAM, then?

9. How was "total uncertainty" derived? In what way does it differ from parameter uncertainty? How does it affect SWI uncertainties (seemed to be one aim of the study)?

Lastly, I encourage the authors to use comments in the code provided on GitHub and used to generate results, in order to make it more understandable.

Here my many minor comments, organized in a list for efficiency:

| Comment | Lines / Table / Figure |
|---|---|
| Reference missing | L15-16 |
| Wrong reference | L31 (wrong year, use 2017) |
| The language needs corrections | L75, 100, 168-170, 187-188, 201-202, 205, 214 |
| Ambiguous or vague statement | L109-113, 151-152, 201-202 |
| Verb missing | L18-24 |
| Scale missing or ambiguous | Fig1b, Fig 9 |
| Units missing | Fig1a, L127-128, Fig4 |
| Axis names missing or ambiguous | Fig2, Fig4-5-6, Fig9 |
| Date ticks missing | Fig2, Fig8 |
| Captions not explicit enough | All figures |
| Errors in the caption | Table 6 |
| Details missing on the methods | L85-87, L109-113, L172-173 |
| Clarifications needed | Eq1, L94-98, 114-116, 151-152 |
| Undefined symbol | Tab2 (ETR) |
| Reformulation needed | L114-116, 134, 164, 181-182, 184-186 |
| Symbol already used | L170 (T) |
| Wrong symbol | Eq12 (t) |
| Repetitive sentences | 193-196 |

References:

Legates, D. R., McCabe, G. J.: Evaluating the use of "goodness-of-fit" measures in hydrologic and hydroclimatic model validation, Water Resources Research, 35, 233-241, 1999.

Loritz, R., Hassler, S. K., Jackisch, C., Allroggen, N., van Schaik, L., Wienhöfer, J., and Zehe, E.: Picturing and modeling catchments by representative hillslopes, Hydrol. Earth Syst. Sci., 21, 1225–1249, https://doi.org/10.5194/hess-21-1225-2017, 2017.

Rodriguez, N. B., Pfister, L., Zehe, E., and Klaus, J.: Testing the truncation of travel times with StorAge Selection functions using deuterium and tritium as tracers, Hydrol. Earth Syst. Sci. Discuss., https://doi.org/10.5194/hess-2019-501, in review, 2019.

Zehe, E., Maurer, T., Ihringer, J., and Plate, E.: Modeling water flow and mass transport in a loess catchment, Phys. Chem. Earth Pt. B, 26, 487–507, https://doi.org/10.1016/S1464-1909(01)00041-7, 2001.

---

## Author Comment (AC1) · 5 Jun 2020

We want to thank Anonymous Referee #1 (AR1) for her/his supporting and motivating words as well as for the valuable and positive criticism. First, we accepted all the minor suggestions, all of them will be implemented.

Below are our responses to each of the main comments.

**In addition, the authors make often statements which are hard to find back in the figures or need a bit more support. For example, based on Figure 8, the authors state that there is a satisfactory performance, but Figure 8b seems quite off in my view, so I am not sure if you can make this statement. More importantly, there are many more tensiometers and events, so why not show more (ideally even all) data? This would generalize and support the statements of the authors much more. The same applies to Figure 11, even though the correlation coefficients are reported in Tables 7 and 8, more results can easily be shown. In addition, a more critical look at this figure is probably justified, instead of just looking at the R-squared values. For example, why are the soil moisture values flattened off? And why is the SWI just changing after a soil moisture value of around 0.155 cm3 / cm3? Just looking at the Rsquared value gives a good indication of the linear relation, but it does not say much about, for example, a consistent bias, so I might be good to look a bit further here. Related to this, I also think it is important that the authors assess the spatial correlation between SWI, HAND and TWI a bit more thoroughly, instead of just a linear regression. For example, normalizing them and subtracting the maps from each other will lead to insights where the values match, and where deviations start to occur. This can also easily be done for the interpolated values of soil moisture (Figure 13)**

We agree that more results can be presented to support the claims made by us in the article. We can add graphs with all the data used in the analysis. We can also analyze the spatial correlation between the SWI, HAND and TWI indices in more detail, instead of linear regression, as suggested by the reviewer.

**Regarding the calibration, what is actually the rational behind using the DREAM algorithm? First, I am a bit confused on how it was implemented, there is a mention of a generalized likelihood function, but in the next sentence (P9.L174) it is stated that the last 7500 samples are used to represent uncertainty. So how was the uncertainty actually represented in the end? In addition, the DREAM method is applied to each event to get parameter estimates, but in a next step, the parameter values of the different events are just averaged, which seems a bit simplistic in contrast to the DREAM algorithm. Why not give more weight to the parameters that are more likely? The added value of this method is also that it gives you uncertainty values, but afterwards, the authors do not really do anything with it in the analyzes of soil moisture indexes. So what is the point of using this method? In addition, I think there is also a strong influence of the chosen evaluation statistics, as the Nash-Sutcliffe efficiency and the RMSE have a strong bias for large values. In other words, in the event based approach in this study, when the height of the peak matches, high values for these metrics are likely to be found. The authors also never specifically mention how they deal with the initial states, which will have a strong influence on the results. Is it correct that these are calibrated?**

We agree that the uncertainty analysis can be better detailed in the sections of Methods and Results. In this work we use a generalized likelihood function proposed by Schoups and Vrugt (2010), which relaxes the commonly assumed premises on residual errors. We do not detail the results regarding the parameter uncertainty, nor the analysis of residual errors. This occurred because initially we did not want to take the focus away from the main analysis, which is the presentation of the SWI index as a parameter representative of the variability of soil moisture over time. In fact, given the difficulty in representing the real uncertainty inherent in the calculation of the SWI, we consider removing the uncertainty analisys from the work, and simply use a standard calibration method as a genetic algorithm; we can even do this if the reviewers find it more consistent. But we can include one more item in the results section showing the uncertainty of the model parameters in the flow generation. As for the initial conditions of the model, these are considered parameters (S1I and S2I), and were obtained through calibration.

**Lastly, I also wonder if it is really a surprise that TWI, HAND and SWI show good correlations. In the end, they are all based on the flow direction map, and especially on the event time-scale for a small basin, I think they should show similar patterns. I am not sure about this, but I just wonder what the authors thoughts are here. Would it not be much more interesting to see if these findings still hold for a longer time-scale and / or even a larger area?**

The three indices are actually based on a map of flow directions, showing similar patterns for this studied basin. The SWI has the particularity of representing the variability of these patterns over time, and we believe that demonstrating this, even for a small basin, is a promising result. Verifying whether this standard remains in a longer time-scale or a larger area would require field data that we unfortunately do not have.

---

## Author Comment (AC2) · 5 Jun 2020

We want to thank Nicolas Rodriguez for his supporting and motivating words as well as for the valuable and positive criticism. First, we accepted all the minor suggestions, all of them will be implemented.

Below are our responses to each of the main comments.

1. **In this work it is not immediately clear why one should bother with a hydrological model which largely simplifies transport processes in the soils and comes with many sources of uncertainties (P, Q, ET, parameters) instead of using simple topographically-derived or precipitation-derived proxies of soil moisture. Why use a model if simpler metrics are satisfactory (especially when keeping table 10 in mind)?**

2. **Given the relatively small size of the catchment (6 ha), the assumption of spatial homogeneity of inputs / land use / soil water parameters, wouldn't a physically-based distributed model (e.g. CATFLOW, Zehe et al., 2001) be more appropriate to simulate soil moisture? This question arises specifically after such models are mentioned in the introduction (L30-36) but not discussed further. In addition, physically-based distributed models would allow a direct and meaningful comparison of simulated vs observed soil moisture.**

The use of the hydrological model and an index derived from it, SWI, opens up possibilities for its future application in different areas. If the SWI correlation is proven as a representative index of soil moisture, it can be used in future works that use soil moisture as a parameter for decision-making in alert for floods, drought, etc., using only data easily available (P, Q, ET) than physical soil parameters, which are difficult to obtain.

3. **The equations could be re-written in order to avoid current ambiguities: • Using subscripts and superscripts for the indexes (0, 1, 2…) • Writing any explicit dependence on time t with (t), and avoiding using a subscript for this • Replacing t-1 by t-Δt • Correctly placing the bound variables (i, j, c) below the sums and giving the lower and upper bounds • Avoiding using a dot symbol between bound variables and elements in the sum (this will be confused with a product) • Correcting equations 7 and 8; Δt is missing to go from fluxes (L/T or L3 /T) to water amounts (L or L3 )**

Ok, the equations will be re-written for better understanding.

4. **Would it not make much more sense to calibrate the DTM rather than transferring the parameters of the calibrated TM to DTM? In the end, the procedure would be the same, but the DTM would be used in the DREAM algorithm instead of TM. Seeing TM working better than DTM in both calibration and validation (tables 5 & 6) suggests that it should have been done this way. As a result, the derived uncertainties are all coming from the calibration of TM, not from DTM which is the model actually used to derive SWI. Thus, the true uncertainties which will affect SWI estimations are not shown.**

We agree that it would be interesting to calibrate the DTM, instead of transferring the calibrated parameters, and this is one of our goals for future works. This procedure requires a

computational effort for which we are not ready yet. This transfer of parameters was made in other previous studies that used different versions of the distributed Tank Model. We agree that the real uncertainties affecting the SWI are not being represented.

**5. The models are calibrated to hydrographs and not soil moisture which is the target. Very good model fits to hydrographs may not guarantee very good fits to soil moisture (e.g. in a catchment dominated by groundwater responses, Loritz et al., 2017; Rodriguez et al., 2019). Is it not possible to similarly calibrate and validate DTM, but to soil moisture observations instead?**

We do not believe it is possible to calibrate the Tank Model or DTM using soil moisture as a parameter without completely modifying the models' equations, proposing a new, physically based model. Lara and Kobiyama (2009) proposed the PM-Tank Model, a physically based version of the Tank Model, which cannot be used in this work, as it requires other physical parameters that we do not have. In this work, the model guaranteed a good fit to the hydrographs, and when we compared the SWI calculated from these with the soil moisture values, there was a good correlation, which for us is an indication of good performance.

**6. The visual comparisons for soil moisture are misleading. Why show only 2 locations out of 9? Is it really meaningful to compare SWI and soil moisture directly? Figure 8 suggests that their relative variations differ by orders of magnitude (especially in Fig8b), and that they may be poorly related (especially in Fig11, the step-wise behavior). The use of a correlation coefficient does not take additive and multiplicative biases into account (Legates and McCabe, 1999), so I suggest to use another performance measure. Also, the correlation coefficient does not allow a spatially-distributed comparison of the results with the observations; while it seems important for the aims of this study (see useful suggestion from first reviewer)**

We agree with this change, we will do the analysis taking into account the bias, using the suggestion of the first reviewer.

**7. How were parameter sets chosen (L192)? I think that the use of an average parameter from all calibration events is not a standard method. In addition, no verification was done in order to check that the mean parameter set actually works well for the calibration events. In my opinion, the standard method would be to calibrate the DTM to all 5 events simultaneously, and validate to the 2 last events simultaneously. Shouldn't DTM work well for all events, to be representative of transport processes in the soils?**

The last 7500 samples are used to represent parameter uncertainty for all the 5 events of calibration. After, an average parameter for all calibration events were calculated. Yes, using the average of the parameters is not a usual method. The average of parameters presented in the work was tested for the calibration and validation periods, and represented all of them well. We can add this information in the work to make readers more secure.

**8. Was the Global Likelihood of Schoups and Vrugt really used? Why choose a constant homoscedastic gaussian likelihood error model, which is already available**

**in DREAM, then? 9. How was "total uncertainty" derived? In what way does it differ from parameter uncertainty? How does it affect SWI uncertainties (seemed to be one aim of the study)?**

As we said to the first reviewer, we agree that the uncertainty analysis can be better detailed in terms of methods and results. In this work we used a generalized likelihood function proposed by Schoups and Vrugt (2010), which relaxes the commonly assumed premises on residual errors. We chose this function precisely because it is already easily available in the DREAMzs algorithm. What happened was that we did not detail the results regarding the parameters uncertainty, nor the analysis of residual errors. This happened because initially we did not want to take the focus away from the main analysis, which is the presentation of the SWI index as a parameter of the variability of soil moisture over time.

Even, given the difficulty in representing the real uncertainty inherent in the calculation of the SWI, or the "total uncertainty", we consider removing this step of uncertainty analysis from the work, and simply using a standard calibration method as a genetic algorithm; we can even do this if the reviewers find it more consistent. But we can include one more item in the results section showing the uncertainty of the model parameters in the flow generation, which is directly linked to the SWI calculation, since both are highly correlated.

References

Lara, P. G. de and Kobiyama, M.:Proposta de Modelo Conceitual: PM Tank Model. Revista Brasileira de Recursos Hídricos, v. 17, n. 3, p. 149-161, 2012.

 Lindner, E. A. and Kobiyama, M.: Proposal of Tank Moisture Index to predict floods and droughts in Peixe River watershed, Brazil. IAHS-AISH Publication, v. 331, p. 314-323, 2009.

Schoups, G. and Vrugt, J. A.: A formal likelihood function for parameter and predictive inference of hydrologic models with correlated, heteroscedastic, and non-Gaussian errors, Water Resources Research, 46, https://doi.org/10.1029/2009WR008933, 2010.